# Advocating for Coccidioidomycosis to Be a Reportable Disease Nationwide in the United States and Encouraging Disease Surveillance across North and South America

**DOI:** 10.3390/jof9010083

**Published:** 2023-01-05

**Authors:** Morgan E. Gorris, Karin Ardon-Dryer, Althea Campuzano, Laura R. Castañón-Olivares, Thomas E. Gill, Andrew Greene, Chiung-Yu Hung, Kimberly A. Kaufeld, Mark Lacy, Edith Sánchez-Paredes

**Affiliations:** 1Information Systems and Modeling, Los Alamos National Laboratory, Los Alamos, NM 87545, USA; 2Department of Geosciences, Texas Tech University, Lubbock, TX 79409, USA; 3Department of Molecular Microbiology and Immunology, University of Texas at San Antonio, San Antonio, TX 78249, USA; 4Unidad de Micología, Facultad de Medicina, Universidad Nacional Autónoma de México, Mexico City 04510, Mexico; 5Environmental Science and Engineering Program, University of Texas at El Paso, El Paso, TX 79968, USA; 6Department of Earth, Environmental and Resource Sciences, University of Texas at El Paso, El Paso, TX 79968, USA; 7Statistical Sciences, Los Alamos National Laboratory, Los Alamos, NM 87545, USA; 8Infectious Disease, Pediatrics, Internal Medicine, University of New Mexico Health, Albuquerque, NM 87106, USA

**Keywords:** coccidioidomycosis, *Coccidioides*, surveillance, public health

## Abstract

Coccidioidomycosis (Valley fever) has been a known health threat in the United States (US) since the 1930s, though not all states are currently required to report disease cases. Texas, one of the non-reporting states, is an example of where both historical and contemporary scientific evidence define the region as endemic, but we don’t know disease incidence in the state. Mandating coccidioidomycosis as a reportable disease across more US states would increase disease awareness, improve clinical outcomes, and help antifungal drug and vaccine development. It would also increase our understanding of where the disease is endemic and the relationships between environmental conditions and disease cases. This is true for other nations in North and South America that are also likely endemic for coccidioidomycosis, especially Mexico. This commentary advocates for US state and territory epidemiologists to define coccidioidomycosis as a reportable disease and encourages disease surveillance in other endemic regions across North and South America in order to protect human health and reduce disease burden.

## 1. Introduction

Coccidioidomycosis, commonly known as Valley fever, is an infectious fungal disease that has gained attention in the United States (US) due to increasing case counts [1]. Humans can contract coccidioidomycosis when they inhale *Coccidioides* spp. fungal spores. It is estimated that 40% of people who inhale the fungus will become symptomatic, with initial symptoms presenting similar to other causes of pneumonia like bacterial and viral infections [2,3]. Approximately 1–5% of coccidioidomycosis patients have severe disseminated forms of the disease spreading from the lung to the skin, lymph nodes, bone, joints, and/or central nervous system, which can lead to life-threatening meningitis [4]. Approximately 200 people in the US die each year from coccidioidomycosis [1]. Coccidioidomycosis can infect a variety of animals, and has been reported in wildlife (e.g., rodents, armadillos, mountain lions, dolphins, sea lions); livestock (e.g., pigs, cattle, sheep, and llamas); companion animals (e.g., dogs, cats, horses); and non-native animals in captivity (e.g., primates, tigers, rhinoceros, kangaroo) [5]. There are currently two species of *Coccidioides* defined: *C. immitis* and *C. posadasii*, though no clinical differences have been noted between the species [6]. Due to underreporting and under-ascertainment, the true number of coccidioidomycosis cases is thought to be much higher [1]. Moreover, the absence of nationwide coccidioidomycosis surveillance limits our understanding of true disease burden and the progress in studying other fundamental research questions, such as the geographical extent of *Coccidioides*.

Though coccidioidomycosis has been a known health threat in the US since the 1930s, not all states are currently report disease cases [7,8]. From 1995–2009 and 2011-current, coccidioidomycosis has been defined in the US as a nationally notifiable disease by the US Centers for Disease Control and Prevention (CDC) and the Council of State and Territorial Epidemiologists (CSTE) [9]. A notifiable disease means there is a standardized case definition and health departments voluntarily submit case count data to the CDC’s National Notifiable Disease Surveillance System; it does not mandate health departments to report the disease. Federally funded facilities and tribal governments are not required to report cases to local health agencies, either [10]. Serological testing for coccidioidomycosis is primarily performed in large commercial laboratories that automatically report laboratory-positive cases to public health agencies, so cases in these communities may still be reported. Some local lawmakers and state health departments have gone a step further to define coccidioidomycosis as a reportable disease in their state; this requires healthcare professionals and laboratories to notify public health departments of cases that meet the coccidioidomycosis CSTE case definition [8]. Currently, 26 US states and the District of Columbia are required to report coccidioidomycosis cases (Figure 1), though the states reporting likely doesn’t encompass the current estimated endemic area.

The CDC and other researchers estimate the area endemic to coccidioidomycosis stretches across the arid to semi-arid portions of the western US [7,8,11]. In the US, the CDC currently estimates *Coccidioides* may live in 12 states (Arizona, California, Colorado, Idaho, New Mexico, Nevada, Oklahoma, Oregon, Texas, Utah, Washington, and Wyoming), but highlights the potential range across 5 more states in the western US (Kansas, Montana, Nebraska, North Dakota, and South Dakota) [7]. A recently developed climate-constrained niche model also estimates 12 states may currently be endemic, but differs slightly from the CDC estimate of where *Coccidioides* may live in that it defines a portion of Kansas and the extreme southwest corner of Nebraska as endemic [11]. Both estimates consider *Coccidioides* to be living in Arizona, California, Colorado, Idaho, Nevada, New Mexico, Oklahoma, Texas, Utah, and Washington. Four states where either the CDC or the climate-constrained niche model estimate *Coccidioides* to be living that do not report coccidioidomycosis cases are Colorado, Idaho, Oklahoma, and Texas. The disease is also endemic to Latin America and cases have been reported in México, Guatemala, Honduras, Colombia, Venezuela, Brazil, Paraguay, Bolivia, and Argentina [12].

Texas is a prime example of where scientific evidence has long supported the state as likely endemic, but coccidioidomycosis is not reportable. Coccidioidomycosis cases have been documented in Texas since the 1930s [13]. Outbreaks of coccidioidomycosis cases were documented in the 1960s in Texas, too: one outbreak in 1965 in El Paso, TX was associated with exposure to a rodent burrow, while a second outbreak was reported in 1966 in Beeville, TX [14,15]. A historical skin test study from the 1950s indicating exposure to *Coccidioides* indicated parts of western Texas to be just as endemic as Kern County, California, with some areas in western Texas presenting with 50–70% of people reacting positive to coccidioidin [16]. Later, one of the first *Coccidioides* niche maps highlighted portions of western Texas as suitable for the fungi [17] and a recently developed habitat suitability map shows much of the western half of Texas as moderate to highly favorable for *Coccidioides* growth [18]. Recent disease surveillance studies support Texas as endemic, though it is difficult to estimate the severity of endemicity compared to other endemic states [19,20]. Despite this information, coccidioidomycosis is currently reported only by El Paso County, the westernmost county in Texas.

The absence of coccidioidomycosis case reporting in Texas, and other likely endemic areas in North and South America, threatens public health by reducing awareness of this disease among the community and physicians, likely leading to a higher disease burden. This commentary advocates for all US states, especially those in or nearby current endemic regions, to define coccidioidomycosis as a reportable disease and encourages disease surveillance across other endemic regions in North and South America. We exemplify the need for reporting coccidioidomycosis in other endemic regions outside of the US by examining the history of case reporting in Mexico. This is a call to action to use our scientific knowledge of coccidioidomycosis and *Coccidioides* in order to mitigate the national and international health impacts of this disease. Doing so will help advance the science and outstanding questions on coccidioidomycosis, including shortening the time to diagnosis, improving antifungal drug and vaccine development, determining where the disease is endemic, identifying the environmental drivers of interannual disease variability, and understanding the severity of endemicity outside the US.

## 2. Clinical Benefits of Coccidioidomycosis as a Reportable Disease

One of the most important public health mechanisms for recognizing, managing, and preventing infectious diseases entails surveillance and reporting of cases. Disease surveillance can increase disease awareness among physicians, which can reduce the time to diagnosis [21]. It is a well-recognized phenomenon in clinical medicine that diseases recognized early have a better chance of good outcome than those recognized belatedly [22]. This is true for coccidioidomycosis [23]. In the highly endemic region of Tucson, Arizona, over 40% of patients with coccidioidomycosis had a diagnosis delay >1 month and many delays resulted in unnecessary courses of antibiotics [23]. The median time among 276 patients between onset of symptoms and coccidioidomycosis diagnosis in this highly endemic region was 23 days [23]. This substantial delay is even greater in non-endemic areas. In an enhanced surveillance study in 14 states outside of the highest endemic regions or in areas considered not endemic (Louisiana, Michigan, Minnesota, Missouri, Montana, Nevada, New Mexico, North Dakota, Ohio, Oregon, Pennsylvania, Utah, Wisconsin, and Wyoming), the median time among 186 patients between seeking healthcare and coccidioidomycosis diagnosis was roughly two weeks longer (38 days), with an incredible range from 1–1654 days [24]. 70% of the patients were originally misdiagnosed and 84% of patients were incorrectly prescribed antibiotics [24]. In a study of coccidioidomycosis case outcomes from 2006–2015 in Utah, approximately one-third of patients were diagnosed with coccidioidomycosis as a workup for malignancy [25], suggesting there should be additional awareness efforts for diagnosing coccidioidomycosis. A first-step action for increasing disease awareness, shortening the time to diagnosis, avoiding unnecessary courses of antibiotics, and avoiding the costs associated with these outcomes is mandatory disease reporting.

Coccidioidomycosis is also endemic to specific regions of the US that are frequently a destination for travel, especially by older people, and result in travel-associated cases [26]. Disease awareness through reporting may help clinicians, especially those practicing outside of endemic areas, consider coccidioidomycosis in their differential diagnosis if they have some understanding of disease prevalence [21]. Prevalence data are produced when there is a reporting mechanism in place.

It is logistically impossible to require the reporting of every infectious disease, so what determines which conditions are reportable? In general, the main considerations include the severity of disease, the cost of managing the disease, acquisition and transmission mechanisms, and what measures are available to mitigate risk.

Persons most at risk for coccidioidomycosis include those with underlying co-morbid disease, such as diabetes, and immunosuppressed patients, particularly those after bone or organ transplantation, those on immunomodulating therapy for autoimmune diseases, and persons with advanced HIV. Pregnancy may also pose increased risk of disease [27]. The proportion of the population with these conditions is increasing, which means the number of persons at risk for severe coccidioidomycosis is also on the rise [28]. Apart from host status, severe disease does occur in people with no recognized immunocompromising disorder, too [29]. The potential for severe outcomes, especially among the increasing numbers of vulnerable and immunocompromised populations, is amplified when the diagnosis is delayed.

For a 2000–2015 baseline time period, the estimated annual medical costs, lost income, and economic welfare losses for coccidioidomycosis in the US were USD 400,000 per case, and the annual average total cost was USD 3.9 billion per year [30]. For 10,359 patients diagnosed in Arizona in 2019, a University of Arizona Health Sciences study estimated total lifetime costs at USD 736 million [31]. That estimate did not include non-hospital medical costs, demands on limited primary care resources, loss of work, and the potential for long term co-morbidities associated with this disease. These cost estimates highlight the impact of the disease on the US economy and the community, especially the highly endemic states such as Arizona.

Since the primary transmission pathway for *Coccidioides* is in the natural environment via soil disturbance and dust, there is relatively little that can be done to mitigate the risk of this disease. The California Division of Occupational Safety and Health (Cal/OSHA) lists numerous recommendations such as training workers on the risk of coccidioidomycosis, limiting worker exposure to dust, wearing respiratory protection with particulate filters rated as N95, N99, N100, P100, or HEPA, and reporting potential symptoms of coccidioidomycosis immediately to a supervisor [32]. For dust mitigation, Cal/OSHA recommends actions such as minimizing disturbing soil, using water or a soil stabilizer to limit dust, suspending work during high winds, and providing enclosed air-conditioned cabs for vehicles that generate dust [32]. The CDC informational webpage on coccidioidomycosis prevention acknowledges that, “it’s very difficult to avoid breathing in the fungus *Coccidioides* in areas where it’s common in the environment” [33].

From the combination of increasing morbidity and mortality from coccidioidomycosis, the high economic and health impact of the disease, and limited mechanisms to mitigate the risk of contracting coccidioidomycosis, it is a compelling case for all US states to define coccidioidomycosis as reportable disease for the clinical benefits alone. The majority of these benefits extend to the veterinary sector, too. Clinical animal cases are likely highly underreported and limited knowledge of the potential for locally acquired infection may delay diagnosis [34]. Sharing surveillance data between veterinary and human medicine could help raise disease awareness, especially since animals can act as early indicators for a biothreat [35].

## 3. Understanding the Public Health Impact of Valley Fever Can Better Justify Antifungal Drug and Vaccine Development

Patients can benefit from an antifungal and/or other immune therapeutics before the onset of disseminated coccidioidomycosis, so early diagnosis of this disease is crucial. Coccidioidomycosis cases are currently confirmed as positive by serology tests. Diagnostic methods of coccidioidomycosis and clinical practice have been well reviewed [36]. Human genetic variations can contribute to high incidence rates of disseminated disease among specific populations, such as African Americans, Asian/Pacific Islanders, and Latinos, compared to European Americans [27,37,38,39]. Furthermore, allelic variations of immune factors of the IL-12-IFN-γ signaling, innate immune sensing, NFκB signaling, and IL-17 signaling molecules have been predicted to associate with the severity of coccidioidomycosis [40,41]. Altogether, those genetic correlation studies suggest that nationwide surveillance may facilitate the rapid identification of coccidioidomycosis patients with any at-risk conditions.

A more complete estimate of the national burden of coccidioidomycosis through nationwide surveillance would further justify the development of novel antifungal drugs. The Infectious Diseases Society of America clinical practice guidelines recommend antifungal chemotherapy for patients at risk for complications, such as moderate to severe pulmonary or disseminated disease [42]. Fluconazole, triazole compounds, and Amphotericin B are the most frequently prescribed antifungal agents for long-term and short-term treatment of moderate to disseminated disease. Fluconazole interacts with 14-demethylase, a cytochrome P450 enzyme that is responsible for catalyzing the conversion of lanosterol to ergosterol, thereby disrupting the stability of the fungal cell membrane [43]. Surprisingly, increased resistance to fluconazole (>16 µg/mL) is documented in >37% of clinical *Coccidioides* isolates [44]. The parasitic lifecycle of *Coccidioides* is unique amongst the medically important fungi as they grow into multinucleated spherules (estimated to be over 1000 nuclei). The fungal cell wall is surrounded by layers of the spherule outer wall (SOW) that is composed of >60% phospholipids/lipids [45,46]. Perhaps the unique cell wall structure and chemical components partly contribute to fluconazole drug resistance. However, it remains unclear how *Coccidioides* resistance to fluconazole correlates to clinical therapeutic outcomes. Amphotericin B is an effective antifungal, but the acute and long-term nephrotoxicity and hepatotoxicity raise concerns. Additionally, there are several promising drug candidates in late-stage clinical development [47,48]. Current drug screening efforts employ a repurposing strategy to identify clinically approved drugs with novel antifungal activity [49]. It is estimated that the cost of drug development from research to clinical use can take USD 314 million to USD 2.8 billion [50]. The screening of repurposing drug libraries can shorten the development time, and it also reduces the cost. Surveillance of coccidioidomycosis cases can provide information on the patient market needs and serve as better justification for anti-*Coccidioides* drug development.

Surveillance data showing the steady rise of coccidioidomycosis cases support the need to develop prophylactic and immunotherapeutic vaccines to manage this disease. Prophylactic vaccines can be administered to naïve individuals who have yet encountered *Coccidioides* to prevent contracting coccidioidomycosis. Immunotherapeutic vaccination can be deployed to coccidioidomycosis patients to enhance their immune responses against this disease. Copious data justify that a vaccine against coccidioidomycosis is feasible, as those who recover from coccidioidomycosis often develop life-long protective immunity [51]. The National Institute of Allergy and Infectious Diseases (NIAID) has set forth a strategic plan to have a human vaccine for coccidioidomycosis undergo clinical trials by 2030 [52]. A whole cell vaccine using formalin-killed spherules (FKS) has been evaluated in a clinical phase III trial against coccidioidomycosis. However, the development of FKS for clinical use was halted due to lack of protective efficacy [53]. Live-attenuated ∆T and the ∆*CPS1* vaccines have been created and both of those vaccines show great protective efficacies against pulmonary coccidioidomycosis in animal models [54,55,56,57]. The live attenuated vaccines are excellent tools for exploring the immune protective mechanisms against *Coccidioides* and they may be developed into veterinary vaccines. Despite the apparent ability of a live-attenuated vaccine to elicit and maintain long-term protective memory responses to *Coccidioides* infection, it poses significant concerns for individuals with underlying conditions of compromised cell-mediated immune systems. Generation of a subunit vaccine using recombinant proteins or mRNA molecules of *Coccidioides* antigens are a safer strategy in the design of a clinically acceptable reagent. Most subunit vaccines are chemically defined and pure. Many investigators have focused on identifying *Coccidioides* antigens which can be used in the development of subunit vaccines [58,59,60]. Additional critical barriers to progress towards the development of a subunit vaccine against coccidioidomycosis are (i) understanding the fundamental knowledge of protective immunity against *Coccidioides* infection; (ii) obtaining protective efficacy and safety data; (iii) reducing the cost by developing a multivalent vaccine that can confer protection against both *C. posadasii* and *C. immitis;* (iv) establishing a translational vaccine platform that can be easily scaled up for vaccine production; and (v) identifying who should get vaccinated for coccidioidomycosis. Nationwide disease surveillance would help identify novel risk factors, describing specific eligibility criteria for an effective design of future vaccine and drug clinical trials. Furthermore, determining fungal susceptibility would allow clinicians to decide the use of prophylactic or immunotherapeutic vaccination for patients based on their history of *Coccidioides*.

## 4. Delineating the Environmental Drivers of *Coccidioides* and Coccidioidomycosis Disease Dynamics and an Estimate of the Current Endemic Area

Reporting coccidioidomycosis cases would help lead to a clearer delineation of the endemic area for this disease, any emergent changes in endemicity, and insight on the environmental conditions that support the endemic area. Comprehensive soil sampling campaigns for the presence of *Coccidioides* are time and labor-intensive, and unless such campaigns are extremely intensive, they may be unreliable due to the patchy distribution of *Coccidioides* in soil [61]. Disease surveillance acts as a proxy for the presence of the fungi and the transmissibility of the fungi to humans. Disease surveillance would be most beneficial in areas where climate or soil conditions are estimated to currently permit endemicity, yet the state or region is not currently reporting [11,18]. In the US, Texas is a large state with basic climate patterns of annual mean temperatures increasing from north to south and precipitation increasing from west to east [62]. These patterns offer a natural experiment in analyzing how coccidioidomycosis case counts, and by proxy the presence of *Coccidioides*, vary with climate conditions. Disease surveillance across the Americas as a whole could help define differing environmental controls on *Coccidioides* and coccidioidomycosis disease dynamics, especially in light of different *Coccidioides* species.

Disease surveillance may act as an early indicator of the shift in the geographical extent of *Coccidioides* and spread of coccidioidomycosis in response to climate change. As temperatures warm and precipitation levels stay dry in the western US, the endemic area for coccidioidomycosis is expected to expand northward [11]. Proactive surveillance in areas projected to be endemic in the future will help raise disease awareness among physicians and track what new communities may be at risk for contracting coccidioidomycosis.

Outside the US, there is less knowledge on the endemic area for coccidioidomycosis, but understanding the environmental controls on US disease cases may provide insight for future studies. We used the methods developed in Gorris et al. (2019) to estimate the regions across North and South America that may be currently endemic to coccidioidomycosis, and therefore, areas we encourage to create disease surveillance programs (Figure 2) [11]. We used mean annual temperature and precipitation from TerraClimate [63] at its native 4 km grid mesh averaged from 2000–2020. We plotted areas in purple that meet the two climate thresholds for endemicity: mean annual temperatures above 10.7 °C and mean annual precipitation levels below 600 mm/year. The resultant endemicity map shows 12 countries as potentially endemic to coccidioidomycosis: the United States, the United Mexican States, Argentina, Bolivia, Brazil, Chile, Colombia, Dominican Republic, Ecuador, Paraguay, Peru, and Venezuela.

Confidence in relating outbreaks of coccidioidomycosis to environmental drivers, like dust storms, requires detailed disease surveillance [64,65]. Although the ability of humans to be infected by *Coccidioides* through wind-eroded dust, and under what conditions it may happen, is subject to debate [65,66], dust exposure has long been hypothesized as a primary mechanism for contracting coccidioidomycosis [67]. A well-known outbreak of coccidioidomycosis occurred in 1977 following a large dust storm originating in the San Joaquin Valley of California [68,69]. The Chihuahuan Desert, an endemic area for *Coccidioides*, experiences some of the highest frequencies of windblown dust in the Western Hemisphere [70], with dust originating from southeastern Arizona across southern New Mexico and far west Texas, as well as from sources in Chihuahua, Mexico [71,72,73,74] (Figure 3). Many dust events originate upwind of Texas and blow into the state, such as from the Chihuahuan Desert regions of New Mexico, Chihuahua (Mexico), and southeastern Arizona [71,72,73,74,75] (Figure 3), and the southwestern Great Plains of eastern New Mexico [76,77]–all hypothesized coccidioidomycosis endemic regions. Dust originating in or upwind of the drylands of western Texas can be transported downwind to the Texas population centers of San Antonio [73], Dallas-Fort Worth [78,79], and Houston [80], and further beyond into neighboring states such as Oklahoma and Kansas [76,78,81,82] (Figure 3) and as far as the northeastern United States and Canada [81,83]. The desert regions of the southwestern US are not only endemic for coccidioidomycosis, but could also cause long-range transport of *Coccidioides* if the fungus remains viable. Enhanced, nationwide coccidioidomycosis surveillance will help us to better understand the mechanisms of population exposure to dust and *Coccidioides* and communicate the risk associated with dust exposure [65].

## 5. Beyond Borders: Reporting Coccidioidomycosis in Mexico

The United Mexican States (Mexico) is another highly endemic area for coccidioidomycosis, though case reporting in the country hasn’t been uniform and it’s thought to be highly underreported. Coccidioidomycosis is the largest systemic fungal infection in Mexico with the first case having been reported in 1945 [12]. Since then, case reporting practices have varied through time. In 1979, the Ministry of Health (MH) of Mexico started weekly announcements of new disease cases. In this registry, coccidioidomycosis cases were only reported from 1989 to 1994 (Figure 4). Starting in 1995, the MH began to operate the National Epidemiological Surveillance System (SINAVE, in Spanish). SINAVE reports 143 diseases subject to conventional surveillance; however, it does not include coccidioidomycosis. Later, in 2000, the MH established a dynamic information consultation service, based on a Multidimensional Online Analytical Processing (MOLAP) technology called Dynamic Cubes, which produces non-obligatory daily reports that record all the diagnoses made by doctors, whether or not they are subject to epidemiological surveillance. Coccidioidomycosis appears again in this database.

It is important to recognize the work of González-Ochoa in the history of coccidioidomycosis reporting, who was a medical mycologist dedicated to studying diseases in the Mexican population, especially in the 1970s and early 1980s. Both González-Ochoa’s extensive research, as well as his administrative position, were instrumental for the recognition of mycotic infections in the Mexican national health system, which probably motivated the extensive recording of cases in the period 1989–1994. From 1989 to 1994, there were 3691 cases of coccidioidomycosis documented in the MH annual reports in Mexico (Figure 4a). Unfortunately, these annual reports do not mention if the diagnosis was made through clinical, microbiology, or immunology tools.

After González-Ochoa’s retirement, for six years, cases of coccidioidomycosis were not recorded. Case reporting began again in 2000. The case data obtained from 2000 to 2019 do not record how the diagnosis was made, but it does show that morbidity decreased by approximately 80% compared to the previous period (1989–1994). Further, the distribution of cases is practically confined to just two states: Sonora and Baja California (Figure 4b). The data are paradoxical because current literature mentions the southern states of the US and the northern states of Mexico as the main global endemic area for coccidioidomycosis. This bias in the data suggests coccidioidomycosis could be a diminishing health concern in Mexico. However, since the clinical symptoms of coccidioidomycosis are very frequently confused with other respiratory tract conditions, there is likely an underreporting of cases in Mexico, which is also supported by the following data:An epidemiological surveillance program has never been implemented to control or prevent this infectionFrom 2010 to 2012, a sub-national survey of intradermal reaction with coccidioidin was carried out in 1081 people from 9 states, and 29.5% of people tested positive for the antigen [84]. The percentage of infection is not negligible, especially if one takes into account that the samples were taken from states with environmental conditions not associated with the presence of *Coccidioides*There are frequent national medical publications of case studies of disseminated coccidioidomycosis in both children and adults [12,85,86], especially in the states of Sonora, Baja California, Coahuila, and Nuevo LeónThe Dynamic Cubes [87] of the National Health System continues to register deaths whose diagnosis is coccidioidomycosis

From inconsistencies in case reporting, it is difficult to draw conclusions on the geographic distribution of coccidioidomycosis in Mexico. Coccidioidin antigen studies carried out in Sonora show that more than 50% of the population react positively [88]. In Baja California, surveys with coccidioidin carried out in 4 municipalities during 2018–2020 showed that 8% of the population is positive for the antigen [89]. In the Dynamic Cubes model, both the place of diagnosis and the origin of the patient are recorded. From these data, we observe that the vast majority of cases recorded in Sonora and Baja California are patients who reside in those same states. Around half of Mexico, especially in northern states, has a dry to very dry climate, conditions that support the presence of coccidioidomycosis throughout the country; however, Chiapas, Oaxaca, and Veracruz are states with large areas of tropical and temperate climates, environments not associated with the presence of *Coccidioides*. For this reason, cases reported in this region are likely where the diagnoses were made, but the fungus is not endemic.

In Mexico, the MH has, because of an established mandate, both the duty to collect, integrate, and disseminate health information necessary for planning, programming, budgeting, and controlling the national health system as well as the duty to know the status and progression of public health. Regrettably, coccidioidomycosis in Mexico is a neglected disease. The quantitative and qualitative concepts that should be used by epidemiologists to study coccidioidomycosis in Mexico to make pertinent decisions are unknown. A more comprehensive method of diagnosing, reporting, collecting, and publishing the case reports of coccidioidomycosis in Mexico would help address these issues, especially delineating the endemic regions in Mexico.

## 6. Discussion

While coccidioidomycosis case counts are already subject to underreporting and under-ascertainment, the lack of uniformity in US states reporting coccidioidomycosis cases, especially those in endemic areas, further exacerbates an underestimation in disease risk. In turn, this underestimation is translated into uncertainties in current research on disease dynamics. Underreporting and under-ascertainment, processes known as imperfect detection, conflate with non-uniform disease surveillance. When creating statistical models of disease counts, some regions may be estimated to have minimal disease counts when they actually could have similar levels of disease with lower rates of reporting. Another issue that arises is that cases can be reported in states where the environmental risk is very low. Oftentimes, this is associated with people that have traveled to a region with high coccidioidomycosis presence and are diagnosed after traveling. This can lead to inflated disease rates in areas where *Coccidioides* is less likely to exist. Eliminating the complications that arise from a lack of uniform disease surveillance will help improve our models of disease burden of coccidioidomycosis.

Of course, there are logistical hurdles for each state defining coccidioidomycosis as a reportable disease. The biggest hurdle is the time and energy of the reporting provider. First, the list of reportable diseases is long and may be challenging for providers to remember which diseases they must report. Second, submitting the appropriate provider report requires time and energy. However, many of the laboratory confirmed tests for coccidioidomycosis automatically send to public health agencies, eliminating the need for providers to submit paperwork. Also, the providers that are more likely to diagnose a coccidioidomycosis case are likely well versed on how to report cases since they are likely dealing with other reportable infectious diseases like tuberculosis.

We return again to discuss the case of Texas, where incidence of coccidioidomycosis in the state is unknown since it’s not a reportable disease, but numerous lines of evidence support Texas as a highly endemic area. Almost 9% of the US population lives in Texas and the state accounts for nearly a third of population growth in the country [90]. Based on historical population growth, in 2022 the population of Texas will be approaching 30 million people [91]. If one assumes the incidence rate in Texas is only 10% of that reported in Arizona during 2019 (14 cases per 100,000 population, a very conservative estimate) [1,92], then approximately 4200 cases of coccidioidomycosis may have been reported in Texas. In another comparison, since historical coccidioidin reactivity rates in western Texas were similar to Kern County, CA, and one assumes the incidence rate of Texas is 10% that of Kern County in 2019 (37 cases per 100,000 population), then approximately 11,100 cases of coccidioidomycosis may have been reported in Texas. The total number of reported coccidioidomycosis cases across states reporting in 2019 was 20,003, so implementing disease surveillance in Texas could very well drastically increase the number of reported cases in the US as a whole.

In addition to population growth, there is evidence certain occupations, such as workers in the oil and gas industry, a growth sector in Texas, may be at increased risk of infection [20]. Conversely, brucellosis and babesiosis ARE reportable diseases in Texas. In 2018, there were 18 cases of brucellosis and 2 cases of babesiosis reported [93]. Both of these conditions are generally more innocuous than coccidioidomycosis in terms of morbidity and mortality, and brucellosis is now nearly eliminated in livestock via a vaccine, so risk to humans is lower as well [94].

Idaho and Colorado are two other states where disease surveillance would be beneficial to understand if there is local acquisition of coccidioidomycosis. Though Idaho is generally considered to be far outside of the traditionally recognized endemic area of the southwestern US, environmental conditions in areas throughout the state are similar to other endemic regions for *Coccidioides*. *Coccidioides* has been isolated over six years in a nearby area in southeastern Washington State [95,96]. The higher resolution climate-constrained niche model for *Coccidioides* (Figure 2) indicates areas extending eastward into Idaho from the endemic region in southeastern Washington State may be suitable for *Coccidioides*, as well as areas in the western Snake River Plain. This includes Boise, the most populous city in Idaho. In Colorado, almost a quarter of the state is highlighted as potentially suitable for *Coccidioides* in the climate-constrained niche model, mostly in the plains area east of the Rocky Mountains (Figure 2). This area is abundant in grass and pastureland [97] and if cattle or other livestock heavily graze, there can be increased soil erosion [98] which could lead to exposure of dust-borne *Coccidioides* spores.

Since population growth in endemic regions of the USA is high, more US states reporting coccidioidomycosis cases is not only important from a healthcare and economic standpoint, it is the “right thing to do” to equip clinicians as they care for patients. As our climate changes and the potential endemic region for coccidioidomycosis expands northward, surveillance programs will help track the geographical expansion of the disease and alert naïve communities when they are at risk of contracting coccidioidomycosis. Understanding the burden of case counts will help support antifungal drug and vaccine development, especially as new populations become exposed to *Coccidioides*. Using our scientific knowledge of coccidioidomycosis and recognizing the numerous benefits in reporting cases, we advocate that coccidioidomycosis becomes a reportable condition in the US and highly recommend surveillance programs in other endemic areas outside of the US to reduce the health effects from coccidioidomycosis.

## Figures and Tables

**Figure 1 jof-09-00083-f001:**
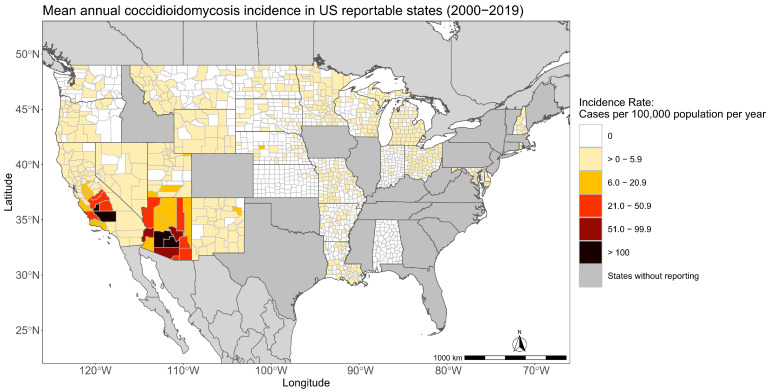
States in the US where coccidioidomycosis is a reportable disease have case incidence in color. Coccidioidomycosis is not reportable in gray states.

**Figure 2 jof-09-00083-f002:**
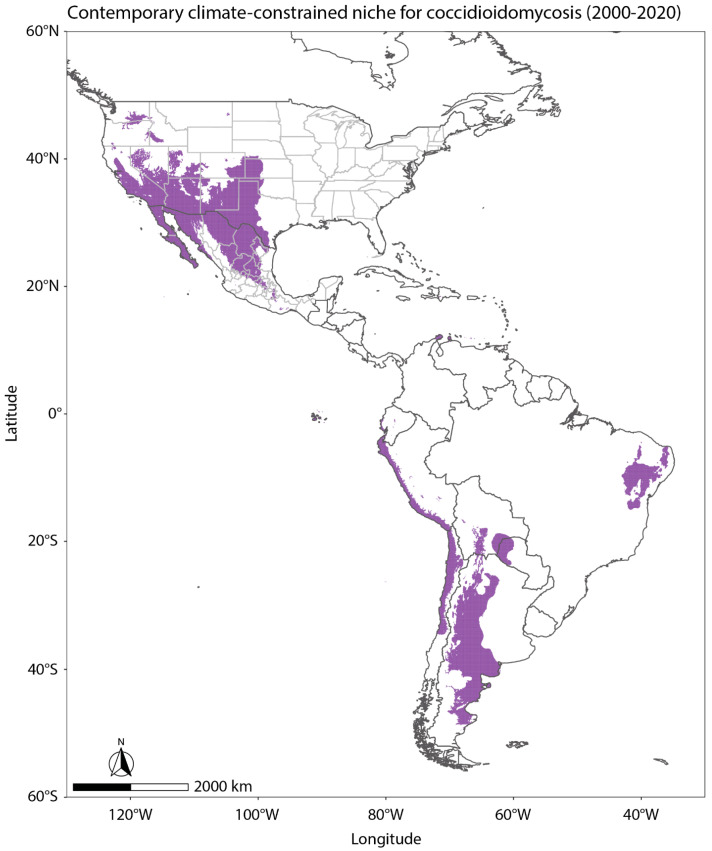
Using the methods from Gorris et al. (2019) [11] and mean annual temperature and precipitation from TerraClimate [63] averaged from 2000–2020, areas plotted in purple meet the two climate thresholds for endemicity: mean annual temperature above 10.7 °C and mean annual precipitation below 600 mm/year. The resultant endemicity map shows 12 countries as potentially endemic to coccidioidomycosis: the United States, the United Mexican States, Argentina, Bolivia, Brazil, Chile, Colombia, Dominican Republic, Ecuador, Paraguay, Peru, and Venezuela.

**Figure 3 jof-09-00083-f003:**
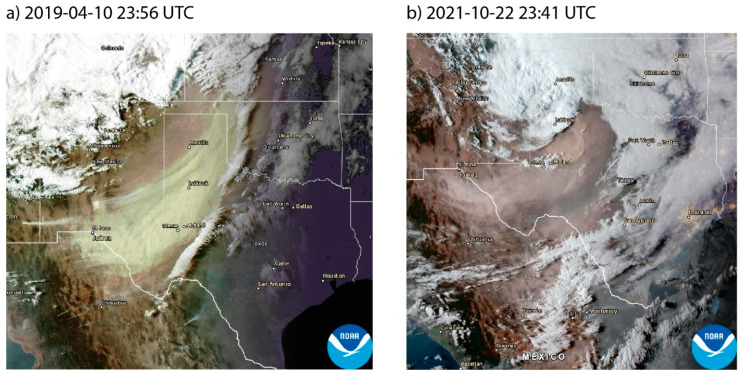
Two examples of recent dust events in Texas and the surrounding region on (**a**) 10 April 2019 at 23:56 UTC and (**b**) 22 October 2021 at 23:41 UTC, captured by the GOES-16 satellite and obtained from AerosolWatch (https://www.star.nesdis.noaa.gov/smcd/spb/aq/AerosolWatch/, accessed on 21 August 2022), showing dust transport from Mexico into Texas (**a**,**b**) and from Texas into Oklahoma and Kansas (**a**).

**Figure 4 jof-09-00083-f004:**
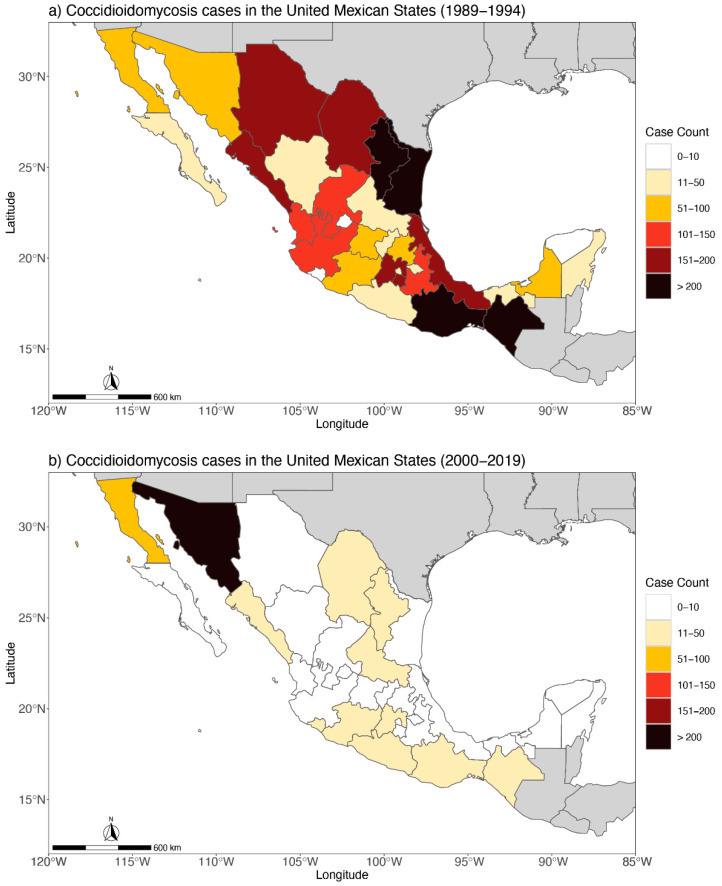
(**a**) Distribution of the 3691 human coccidioidomycosis cases in the United Mexican States from 1989 to 1994 and (**b**) 766 human coccidioidomycosis cases in the United Mexican States from 2000 to 2019.

## Data Availability

Restrictions apply to the availability of these data since they are public health data from the CDC and Ministry of Health of Mexico. Please contact the corresponding author for more information.

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
