# Peer review of "Advocating for Coccidioidomycosis to Be a Reportable Disease Nationwide in the United States and Encouraging Disease Surveillance across North and South America"

_jof, 2023, doi:10.3390/jof9010083_

Round 1
Reviewer 1 Report
The authors have presented a strong and compelling argument on a topic of great importance. It is well-written and supported with robust evidence. Comments and suggestions are provided below:
· The main point to increase case reporting for coccidioidomycosis is well taken. However, the terminology of ‘nationally reportable’ may not be well suited for a public health audience as reportability is inherently determined by the state or territory (https://www.cdc.gov/nndss/about/index.html), and ‘nationally reportable’ may give the connotation that there is a national authority to designate reportability (as is the case with nationally notifiable status). Would suggest rephrasing to advocate for reportable status in a greater number of states (could potentially use ‘nationwide’ as long as it is clear that reportability is determined at the state/territorial level)
· Regarding a vaccine, it may be beneficial to note that surveillance will not only be useful to identify target populations for the vaccine itself but would also be useful as a means to identify candidates for vaccine clinical trials
o Some of the more technical details regarding the vaccine can probably be removed as those points are not as closely tied to case reporting or increased surveillance
· The commentary could be further strengthened by providing a counterpoint (such as the logistical burden of case reporting, particularly with clinical criteria), and then responding to it (such as opportunities to use a laboratory-only case definition per the new CSTE case definition. Footnotes in the new definition demonstrate how states beyond California and Arizona can leverage this classification).
· Lines 33-34: Suggest rephrasing to highlight a resemblance to ‘other respiratory illnesses’ or ‘other pneumonias’ based on findings from Benedict et al., 2021 https://academic.oup.com/cid/article/73/11/e4336/5923374?login=true
· Lines 46-47: Consider changing to “limited national surveillance program” since NNDSS is technically a national surveillance program even though coccidioidomycosis is not reportable in all states.
· Lines 66-67: Reportable in 26 states and the District of Columbia (if counting Washington, where it is reportable as a rare disease of public health significance) https://www.cdc.gov/fungal/fungal-disease-reporting-table.html
· Line 81: Please soften the language to indicate that CDC estimates that Oregon Wyoming may be endemic.
· Line 87-103: While this paragraph is specific to Texas, it may be worth highlighting somewhere in the commentary that the arid region where cocci was identified in Washington State extends into Idaho, and surveillance in Idaho would be beneficial to understand potential local acquisition in the state.
· Lines 123-130: Consider using findings from this enhanced cocci surveillance report that discusses misdiagnoses, antibiotic use, etc. in low/non-endemic states: https://www.ncbi.nlm.nih.gov/pmc/articles/PMC6056093/
· Lines 134-135: Suggest emphasizing clinicians outside of endemic areas
· Lines 208-209: Consider noting that we are unsure how these higher MIC levels correlate to clinical outcomes
· Lines 216-218: Consider mentioning that there are some promising medications coming to the market that have activity against Coccidioides: https://www.ncbi.nlm.nih.gov/pmc/articles/PMC8501344/
Author Response
Point by Point Reviewer Responses for:
jof-2057656
Title: Advocating for coccidioidomycosis to be a reportable disease nationwide in the United States and encouraging disease surveillance across North and
South America
Thank you to the reviewers for your helpful feedback. Please see our point-by-point responses in blue to your suggestions in black.
Reviewer 1
The authors have presented a strong and compelling argument on a topic of great importance. It is well-written and supported with robust evidence. Comments and suggestions are provided below:
- The main point to increase case reporting for coccidioidomycosis is well taken. However, the terminology of ‘nationally reportable’ may not be well suited for a public health audience as reportability is inherently determined by the state or territory (https://www.cdc.gov/nndss/about/index.html), and ‘nationally reportable’ may give the connotation that there is a national authority to designate reportability (as is the case with nationally notifiable status). Would suggest rephrasing to advocate for reportable status in a greater number of states (could potentially use ‘nationwide’ as long as it is clear that reportability is determined at the state/territorial level)
Thank you for bringing up this important point. We changed our title to better reflect this: “Advocating for coccidioidomycosis to be a reportable disease nationwide in the United States and encouraging disease surveillance across North and South America”
We also changed the language in numerous instances throughout the text to better reflect that there is no national authority.
- Regarding a vaccine, it may be beneficial to note that surveillance will not only be useful to identify target populations for the vaccine itself but would also be useful as a means to identify candidates for vaccine clinical trials
- Some of the more technical details regarding the vaccine can probably be removed as those points are not as closely tied to case reporting or increased surveillance
Starting on Line 317, we added: “National disease surveillance would identify novel risk factors, describing specific eligibility criteria for an effective design of future vaccine and drug clinical trials. Furthermore, determining fungal susceptibility would allow clinicians to decide the use of prophylactic or immunotherapeutic vaccination for patients based on their history of Coccidioides.” We consider describing the various hurdles to addressing safe and effective vaccine development necessary and feel it would be a disservice to omit its challenges.
- The commentary could be further strengthened by providing a counterpoint (such as the logistical burden of case reporting, particularly with clinical criteria), and then responding to it (such as opportunities to use a laboratory-only case definition per the new CSTE case definition. Footnotes in the new definition demonstrate how states beyond California and Arizona can leverage this classification).
Thank you for this helpful feedback. We added a paragraph in the Discussion section on line 485 to pose a counterpoint and a response. It reads:
“Of course, there are logistical hurdles for each state defining coccidioidomycosis as a reportable disease. The biggest hurdle is the time and energy of the reporting provider. First, the list of reportable diseases is long and may be challenging for providers to remember which diseases they must report. Second, submitting the appropriate provider report requires time and energy. However, many of the laboratory confirmed tests for coccidioidomycosis automatically send to public health agencies, eliminating the need for providers to submit paperwork. Also, the providers that are more likely to diagnose a coccidioidomycosis case are likely well versed on how to report cases since they are likely dealing with other reportable infectious diseases like tuberculosis.”
- Lines 33-34: Suggest rephrasing to highlight a resemblance to ‘other respiratory illnesses’ or ‘other pneumonias’ based on findings from Benedict et al., 2021 https://academic.oup.com/cid/article/73/11/e4336/5923374?login=true
Thank you for this suggestion. We rewrote the sentence in the Introduction on line 47, which now reads: “It is estimated that 40% of people who inhale the fungus will become symptomatic, with initial symptoms presenting similar to other causes of pneumonia like bacterial and viral infections (Benedict et al., 2021; Smith et al., 1948).”
We added the following reference:
Benedict, K., Kobayashi, M., Garg, S., Chiller, T., & Jackson, B. R. (2021). Symptoms in Blastomycosis, Coccidioidomycosis, and Histoplasmosis Versus Other Respiratory Illnesses in Commercially Insured Adult Outpatients—United States, 2016–2017. Clinical Infectious Diseases, 73(11), e4336–e4344. https://doi.org/10.1093/cid/ciaa1554
- Lines 46-47: Consider changing to “limited national surveillance program” since NNDSS is technically a national surveillance program even though coccidioidomycosis is not reportable in all states.
We changed the sentence on like 61 to read: “Moreover, the absence of nationwide coccidioidomycosis surveillance limits our understanding of true disease burden and the progress in studying other fundamental research questions, such as the geographical extent of Coccidioides.”
- Lines 66-67: Reportable in 26 states and the District of Columbia (if counting Washington, where it is reportable as a rare disease of public health significance) https://www.cdc.gov/fungal/fungal-disease-reporting-table.html
Thank you for mentioning this. We changed the sentence on line 89 to read:
“Currently, 26 US states and the District of Columbia are required to report coccidioidomycosis cases (Figure 1), though the states reporting likely doesn’t encompass the current estimated endemic area.”
- Line 81: Please soften the language to indicate that CDC estimates that Oregon Wyoming may be endemic.
We changed the language throughout this paragraph starting on line 93 to use the terms on the CDC “More Information” website
We state that “the CDC currently estimates Coccidioides may live in…, but highlights the potential range across 5 more states…”
We deleted the clause in the next sentence that stated the CDC estimates Oregon and Wyoming are endemic to be consistent.
- Line 87-103: While this paragraph is specific to Texas, it may be worth highlighting somewhere in the commentary that the arid region where cocci was identified in Washington State extends into Idaho, and surveillance in Idaho would be beneficial to understand potential local acquisition in the state.
Thank you for bringing up this point. We added a paragraph in the discussion section that also suggests reporting coccidioidomycosis in Idaho and Colorado. We appreciate your suggestion to add the discussion point about the endemic are in Washington state. The paragraph on line 519 reads:
“Idaho and Colorado are two other states where disease surveillance would be beneficial to understand if there is local acquisition of coccidioidomycosis. Though Idaho is generally considered to be far outside of the traditionally recognized endemic area of the southwestern US, environmental conditions in areas throughout the state are similar to other endemic regions for Coccidioides. Coccidioides has been isolated over six years in a nearby area in southeastern Washington State (Chow et al., 2021; Litvintseva et al., 2015). The higher resolution climate-constrained niche model for Coccidioides (Figure 2) indicates areas extending eastward into Idaho from the endemic region in southeastern Washington State may be suitable for Coccidioides, as well as areas in the western Snake River Plain. This includes Boise, the most populous city in Idaho. In Colorado, almost a quarter of the state is highlighted as potentially suitable for Coccidioides in the climate-constrained niche model, mostly in the plains area east of the Rocky Mountains (Figure 2). This area is abundant in grass and pastureland (Han et al., 2012) and if cattle or other livestock heavily graze, there can be increased soil erosion (Horrigan et al., 2002) which could lead to exposure to aerosolized Coccidioides spores.”
- Lines 123-130: Consider using findings from this enhanced cocci surveillance report that discusses misdiagnoses, antibiotic use, etc. in low/non-endemic states: https://www.ncbi.nlm.nih.gov/pmc/articles/PMC6056093/
Thank you for suggesting this great resource. We incorporated this citation into the piece and rewrote the first paragraph of the “Clinical benefits” section starting on line 152. It now reads:
“One of the most important public health mechanisms for recognizing, managing, and preventing infectious diseases entails surveillance and reporting of cases. Disease surveillance can increase disease awareness among physicians, which can reduce the time to diagnosis (Croskerry, 2003). It is a well-recognized phenomenon in clinical medicine that diseases recognized early have a better chance of good outcome than those recognized belatedly (Pruinelli et al., 2018). This is true for coccidioidomycosis (Donovan et al., 2019). In the highly endemic region of Tucson, Arizona, over 40% of patients with coccidioidomycosis had a diagnosis delay >1 month and many delays resulted in unnecessary courses of antibiotics (Donovan et al., 2019). The median time among 276 patients between onset of symptoms and coccidioidomycosis diagnosis in this highly endemic region was 23 days (Donovan et al., 2019). This substantial delay is even greater in non-endemic areas. In an enhanced surveillance study in 14 states outside of the highest endemic regions or in areas considered not endemic (Louisiana, Michigan, Minnesota, Missouri, Montana, Nevada, New Mexico, North Dakota, Ohio, Oregon, Pennsylvania, Utah, Wisconsin, and Wyoming), the median time among 186 patients between seeking healthcare and coccidioidomycosis diagnosis was 38 days, with an incredible range from 1–1,654 days (Benedict et al., 2018). 70% of the patients were originally misdiagnosed and 84% of patients were incorrectly prescribed antibiotics (Benedict et al., 2018). In a study of coccidioidomycosis case outcomes from 2006-2015 in Utah, approximately one-third of patients were diagnosed with coccidioidomycosis as a workup for malignancy (Carey et al., 2021), suggesting there should be additional awareness efforts for diagnosing coccidioidomycosis. A first-step action for increasing disease awareness, shortening the time to diagnosis, avoiding unnecessary courses of antibiotics, and avoiding the costs associated with this outcomes is mandatory disease reporting.”
- Lines 134-135: Suggest emphasizing clinicians outside of endemic areas
Line 180, We changed this to read, “Disease awareness through reporting may help clinicians, especially those practicing outside of endemic areas, consider coccidioidomycosis in their differential diagnosis, especially if they have some understanding of disease prevalence (Croskerry, 2003).”
- Lines 208-209: Consider noting that we are unsure how these higher MIC levels correlate to clinical outcomes
Line 269, we added “However, it remains unclear Coccidioides resistance to fluconazole correlates to clinical therapeutic outcomes.”
- Lines 216-218: Consider mentioning that there are some promising medications coming to the market that have activity against Coccidioides: https://www.ncbi.nlm.nih.gov/pmc/articles/PMC8501344/
We deleted a sentence at the end of the paragraph for flow and added Line 271, “Additionally, there are several promising drug candidates in late-stage clinical development (Hoenigl et al, 2021; Thompson et al., 2019).”
Reviewer 2
The manuscript, “Advocating for coccidioidomycosis as a nationally reportable disease in the United States and encouraging disease surveillance across North and South America” by Gorris et al., is a review of coccidioidomycosis and a call for requiring the disease to be nationally notifiable and how this reporting would have an effect on reducing morbidity and mortality. The manuscript also discusses the apparent expansion of the endemic region for coccidioidomycosis how this data would also be used to track the movement of disease into neighboring states.
Questions and comments
- Lines 73-84: While coccidioidomycosis has be diagnosed in multiple states and may be expanding into other states care must be taken on what states are currently considered endemic regions and which might become endemic in the future. The author uses information from the CDC website which states Arizona, California, Nevada, New Mexico, Texas, and Utah as areas where Coccidioides is considered to “live”. Oregon and Wyoming are not in this list. Line 82 also adds more states (Colorado, Idaho, Oklahoma) than currently considered endemic. The author appears to be basing his list on the CDC maps which do depict a small corner of these states as lightly shaded area showing a potential source but hardly considered endemic. They may become ‘endemic’ in the future if the author’s predictions are realized. This paragraph should be rewritten to more accurately define the current data.
Thank you for raising this concern. In this paragraph, we are addressing what areas may currently be “endemic” to Coccidioides. However, we softened the language to better reflect current knowledge and how the CDC addresses their maps. To do so, we used the “More information on the Valley fever maps…” website:
We changed the language throughout this paragraph starting on line 93 to better reflect the language used on the CDC website (as also suggested by Reviewer 1). For example, we now state that “the CDC currently estimates Coccidioides may live in…, but highlights the potential range across 5 more states…”
- In the discussion, the author spends a lot of time focusing on Texas. Are there other states that the author feels that should also be focused on as well?
Thank you for bringing up this point. We added a paragraph in the discussion section that also suggests reporting coccidioidomycosis in Idaho and Colorado. The paragraph on line 519 reads:
“Idaho and Colorado are two other states where disease surveillance would be beneficial to understand if there is local acquisition of coccidioidomycosis. Though Idaho is generally considered to be far outside of the traditionally recognized endemic area of the southwestern US, environmental conditions in areas throughout the state are similar to other endemic regions for Coccidioides. Coccidioides has been isolated over six years in a nearby area in southeastern Washington State (Chow et al., 2021; Litvintseva et al., 2015). The higher resolution climate-constrained niche model for Coccidioides (Figure 2) indicates areas extending eastward into Idaho from the endemic region in southeastern Washington State may be suitable for Coccidioides, as well as areas in the western Snake River Plain. This includes Boise, the most populous city in Idaho. In Colorado, almost a quarter of the state is highlighted as potentially suitable for Coccidioides in the climate-constrained niche model, mostly in the plains area east of the Rocky Mountains (Figure 2). This area is abundant in grass and pastureland (Han et al., 2012) and if cattle or other livestock heavily graze, there can be increased soil erosion (Horrigan et al., 2002) which could lead to exposure to aerosolized Coccidioides spores.”
Reviewer 2 Report
The manuscript, “Advocating for coccidioidomycosis as a nationally reportable disease in the United States and encouraging disease surveillance across North and South America” by Gorris et al., is a review of coccidioidomycosis and a call for requiring the disease to be nationally notifiable and how this reporting would have an effect on reducing morbidity and mortality. The manuscript also discusses the apparent expansion of the endemic region for coccidioidomycosis how this data would also be used to track the movement of disease into neighboring states.
Questions and comments
Lines 73-84: While coccidioidomycosis has be diagnosed in multiple states and may be expanding into other states care must be taken on what states are currently considered endemic regions and which might become endemic in the future. The author uses information from the CDC website which states Arizona, California, Nevada, New Mexico, Texas, and Utah as areas where Coccidioides is considered to “live”. Oregon and Wyoming are not in this list. Line 82 also adds more states (Colorado, Idaho, Oklahoma) than currently considered endemic. The author appears to be basing his list on the CDC maps which do depict a small corner of these states as lightly shaded area showing a potential source but hardly considered endemic. They may become ‘emdemic’ in the future if the author’s predictions are realized. This paragraph should be rewritten to more accurately define the current data.
In the discussion, the author spends a lot of time focusing on Texas. Are there other states that the author feels that should also be focused on as well?
Author Response

(The authors gave the same response as above.)
